# Upper critical field reaches 90 tesla near the Mott transition in fulleride superconductors

Y. Kasahara[1], Y. Takeuchi[2], R.H. Zadik[3], Y. Takabayashi[4], R.H. Colman[3], R.D. McDonald[5], M.J. Rosseinsky[6], K. Prassides[4,7] & Y. Iwasa[2,8]

Controlled access to the border of the Mott insulating state by variation of control parameters offers exotic electronic states such as anomalous and possibly high-transition-temperature ($T_c$) superconductivity. The alkali-doped fullerides show a transition from a Mott insulator to a superconductor for the first time in three-dimensional materials, but the impact of dimensionality and electron correlation on superconducting properties has remained unclear. Here we show that, near the Mott insulating phase, the upper critical field $H_{c2}$ of the fulleride superconductors reaches values as high as $\sim 90\,T$—the highest among cubic crystals. This is accompanied by a crossover from weak- to strong-coupling superconductivity and appears upon entering the metallic state with the dynamical Jahn–Teller effect as the Mott transition is approached. These results suggest that the cooperative interplay between molecular electronic structure and strong electron correlations plays a key role in realizing robust superconductivity with high-$T_c$ and high-$H_{c2}$.

[1] Department of Physics, Kyoto University, Kyoto 606-8502, Japan. [2] Quantum-Phase Electronics Center (QPEC) and Department of Applied Physics, University of Tokyo, Tokyo 113-8656, Japan. [3] Department of Chemistry, Durham University, Durham DH1 3LE, UK. [4] WPI—Advanced Institute for Materials Research, Tohoku University, Sendai 980-8577, Japan. [5] NHMFL, Los Alamos National Laboratory, Los Alamos, New Mexico 87545, USA. [6] Department of Chemistry, University of Liverpool, Liverpool L69 7ZD, UK. [7] Japan Science and Technology Agency (JST), ERATO Isobe Degenerate π-Integration Project, Tohoku University, Sendai 980-8577, Japan. [8] RIKEN Center for Emergent Matter Science (CEMS), Wako, Saitama 351-0198, Japan. Correspondence and requests for materials should be addressed to Y.K. (email: ykasahara@scphys.kyoto-u.ac.jp) or to Y.I. (email: iwasa@ap.t.u-tokyo.ac.jp).

The interplay between superconductivity and electron correlations is one of the central issues in condensed matter physics. Superconducting (SC) materials based on Mott insulators, such as two-dimensional (2D) cuprates[1] and organic charge-transfer salts[2], are model platforms that have been extensively studied thus far. A dome-like dependence of the SC transition temperature $T_c$ as a function of tuning parameters, such as carrier doping and pressure, has been discussed as a fingerprint of unconventional superconductivity[3]. Recent physical and chemical pressure studies of $Cs_3C_{60}$ have revealed that the family of cubic fullerides $A_3C_{60}$ (A: alkali metal), where superconductivity emerges from the Mott insulating state driven by dynamical intramolecular Jahn–Teller (JT) distortions and strong Coulomb repulsion, is a new example of superconductors that show a dome-like SC phase diagram as a function of unit-cell volume $V$ (refs 4–9). This suggests the importance of strong electron correlation to SC mechanisms[10] and the need for further treatment beyond conventional framework of theory[11]. Recent study has revealed a crossover in the normal state from the conventional Fermi liquid to a nontrivial metallic state where JT distortions persist (JT metal)[9,12]. There, localized electrons coexist with itinerant electrons microscopically and heterogeneously.

The dependence of the upper critical field $H_{c2}$ on $T_c$ is relevant to the understanding of the dome-like SC phase because $H_{c2}$ is determined by the coherence length (the size of the Cooper pair) as well as the strength of the pairing potential. Therefore, $H_{c2}$ is also important to understand the underlying mechanism of the superconductivity. However, for the fullerides, $H_{c2}$ as a function of $V$ has not as yet been determined due to the very large $H_{c2}$ and the need for high pressure to access superconductivity in $Cs_3C_{60}$.

Here we report measurements of $H_{c2}$ using a pulsed magnetic field in $Rb_xCs_{3-x}C_{60}$, where superconductivity appears near the Mott transition even at ambient pressure[9]. In proximity to the Mott transition, $H_{c2}$ is enhanced up to ~90 T, which is the highest among cubic superconductors. We uncovered that $H_{c2}$ and the pairing strength increase concomitantly with increasing lattice volume near the Mott transition, suggesting that molecular characteristics as well as electron correlations play important roles for realizing superconductivity with high-$T_c$ and high-$H_{c2}$ in molecular materials.

## Results

**Temperature dependence of upper critical field.** $H_{c2}$ of the fulleride superconductors (Fig. 1a) $Na_2CsC_{60}$, $K_3C_{60}$, and $Rb_xCs_{3-x}C_{60}$ ($0 < x \leqq 3$), has been measured by a radiofrequency technique in pulsed magnetic fields[13] up to 62 T (see Methods). In $Rb_xCs_{3-x}C_{60}$ with $x \leqq 1$, the dynamical JT distortions (Fig. 1b) persist down to low temperature and coexist with the metallic state, and superconductivity emerges from this JT metal state ($V_{max} < V < V_{cr}$, in Fig. 1c). Figure 2 shows temperature ($T$) variations of frequency shift $\Delta f$ as a function of the magnetic field $H$ for $Rb_xCs_{3-x}C_{60}$ ($x = 2$, 0.75, and 0.35) (see also Supplementary Fig. 1). The $T$ dependence of $H_{c2}$, $H_{c2}(T)$, was determined as a point at which $\Delta f$ intercepts the normal-state

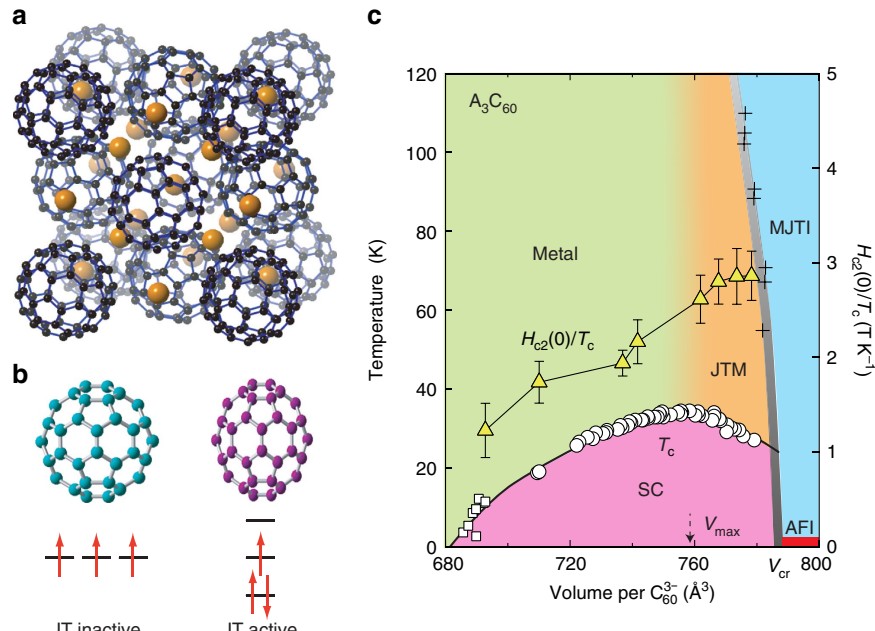

**Figure 1 | Crystal structure and electronic phase diagram of fcc fullerides. (a)** Crystal structure of fcc $A_3C_{60}$. Orange and black spheres represent A and C atoms, respectively. The $C_{60}^{3-}$ anions adopt two orientations related by 90° rotation about the [100] axis. Only one is shown at each site. **(b)** Schematic structures of $C_{60}^{3-}$ anions and molecular $t_{1u}$ orbitals. At low $V$, $C_{60}^{3-}$ anions are isotropic, and $t_{1u}$ orbitals are triply degenerate. At large $V$, dynamical JT distortions give rise to threefold splitting of the $t_{1u}$ orbitals. **(c)** Electronic phase diagram of cubic fullerides. Squares and circles are the superconducting (SC) transition temperature $T_c$ for f.c.c. $C_{60}^{3-}$ anion packings with $Pa\bar{3}$ symmetry and $Fm\bar{3}m$ symmetry, respectively. In fcc-$Cs_3C_{60}$ at ambient pressure, an electron-correlation-driven insulating state (Mott-Jahn–Teller insulator, MJTI) appears, which is accompanied by an intramolecular dynamical Jahn–Teller (JT) effect distorting the $C_{60}^{3-}$ anions and stabilizing the low-spin ($S = 1/2$) states that give rise to an antiferromagnetic insulating (AFI) state at low temperatures. In the metallic regime, gradient shading from green to orange schematically illustrates a crossover from the conventional metal to unusual metallic state where JT distortions persist, which we define as the JT metal (JTM) state. The grey line represents the MJTI-to-JTM crossover line, where the crossover temperatures (crosses) were obtained from X-ray powder diffraction, nuclear magnetic resonance spectroscopy, and infrared spectroscopy[9]. The ratio of upper critical field at $T = 0$ and $T_c$, $H_{c2}(0)/T_c$ (yellow triangles), shows an enhancement in the JTM regime. Error bars represent the s.d. in the values of $H_{c2}(0)$ estimated from the least-squares fits of equation (1) to $H_{c2}(T)$ data.

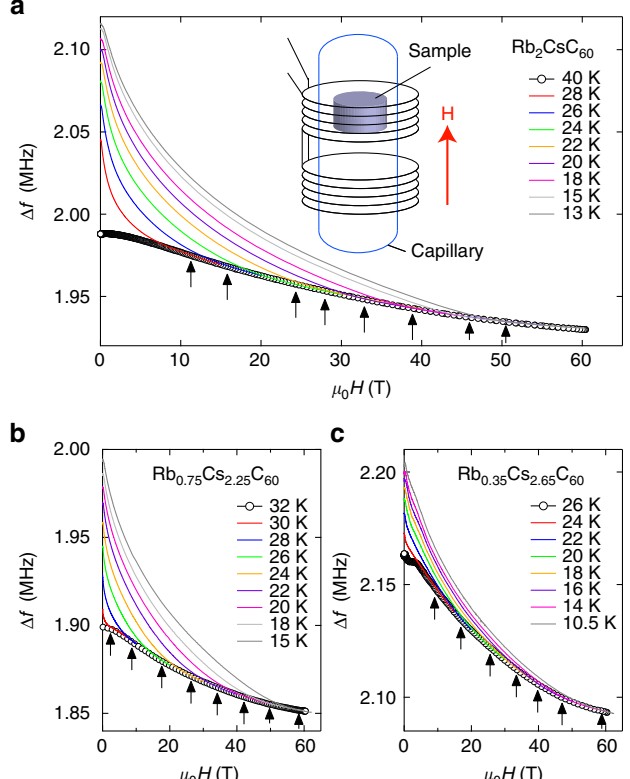

**Figure 2 | Determination of upper critical fields.** Frequency shift ($\Delta f$) as a function of magnetic field for Rb$_x$Cs$_{3-x}$C$_{60}$ with (**a**) $x=2$, (**b**) 0.75, and (**c**) 0.35 at selected temperatures. Open circles are $\Delta f$ taken at $T>T_c$ as a normal-state background signal. The arrows indicate $H_{c2}(T)$ determined from the point deviating from the background signal. Inset in **a** shows a schematic of the experimental set-up. The sample in a capillary was inserted in one coil of the pair wound clockwise and anti-clockwise to compensate induced voltages that are generated during the field pulse.

background (arrows in Fig. 2). $H_{c2}(T)$ curves for A$_3$C$_{60}$ are plotted in Fig. 3a,b for $V \lesssim V_{max}$ and $V_{max} < V < V_{cr}$ in the proximity of the Mott transition, respectively. $H_{c2}(T)$ increases linearly with decreasing $T$ near $T_c$ and has a tendency to saturate at low temperatures. No obvious upturn of $H_{c2}(T)$ is found in any of the samples measured, implying that $H_{c2}(T)$ can be understood within a simple single-band picture despite the multiband nature of the triply degenerate $t_{1u}$ orbitals of C$_{60}^{3-}$ anions, in contrast to MgB$_2$ and iron pnictides where multiband and multigap behaviour with upturn or quasilinear $T$ dependence down to $T \sim 0$ is commonly observed.

**Volume dependence.** In spin-singlet superconductors, $H_{c2}$ is determined by two distinct effects, i.e., the orbital and the Pauli paramagnetic effect. The orbital limit and Pauli limit are given by $H_{c2}^{orb}(0) = 0.69 T_c |dH_{c2}/dT|_{T=T_c} = \Phi_0/2\pi\xi_{GL}^2$ and $H_P = \Delta_0/\sqrt{2}\mu_B$, respectively ($\Phi_0$, $\xi_{GL}$, $\Delta_0$, and $\mu_B$ are the flux quantum, Ginzburg–Landau (GL) coherence length, superconducting gap and Bohr magneton, respectively)[14,15]. In a weak-coupling BCS super-conductor, the Pauli limit is $H_P^{BCS}[T] = 1.84 T_c[K]$. A simple estimation from $H_{c2}^{orb}(0)$ gives $\xi_{GL} = 1.8–4.6$ nm (Supplementary Table 1), which is comparable to the lattice constant. It should be noted that the fulleride superconductors are in the dirty limit, $\ell \lesssim \xi_0$ ($\ell$ and $\xi_0$ are the mean free path and Pippard coherence length, respectively)[16,17], as demonstrated by transport and optical measurements[16,17]. The orientational disorder of the C$_{60}^{3-}$ anions

can account for the short $\ell$, which is comparable to the intermolecular separation. The relation $\xi_{GL} = 0.85\sqrt{\xi_0 \ell}$ in the dirty limit, where $\xi_0 = \hbar v_F/\pi\Delta_0$ and $m^* v_F = \hbar k_F = \hbar \left(\frac{3\pi^2 N}{V}\right)^{1/3}$ ($v_F$, $m^*$, $k_F$, and $N$ are the Fermi velocity, effective mass, Fermi momentum, and number of electrons per C$_{60}$, respectively) for the parabolic band approximation yield $H_{c2}^{orb}(0) = 0.22 \frac{\Phi_0}{\hbar^2} \frac{\Delta_0 m^*}{\ell} \left(\frac{V}{N}\right)^{1/3}$. In the extreme cases ($H_{c2}^{orb} \gg H_P$ or $H_{c2}^{orb} \ll H_P$), $H_{c2}(0)$ is determined solely by $H_{c2}^{orb}$ or $H_P$. However, when these two quantities are comparable, $H_{c2}(T)$ can be described by the extended WHH formula[14], which considers both the orbital and Pauli paramagnetic effects as well as spin–orbit scattering,

$$
\begin{aligned}
\ln\left(\tfrac{1}{t}\right) =\; & \left(\tfrac{1}{2} + \tfrac{i\lambda_{so}}{4\gamma}\right)\psi\left(\tfrac{1}{2} + \tfrac{\bar{h} + \tfrac{1}{2}\lambda_{so} + i\gamma}{2t}\right) \\
& + \left(\tfrac{1}{2} - \tfrac{i\lambda_{so}}{4\gamma}\right)\psi\left(\tfrac{1}{2} + \tfrac{\bar{h} + \tfrac{1}{2}\lambda_{so} - i\gamma}{2t}\right) - \psi\left(\tfrac{1}{2}\right),
\end{aligned} \quad (1)
$$

where $t = T/T_c$, $\bar{h} = 0.281 H_{c2}(T)/H_{c2}^{orb}(0)$, $\gamma = \sqrt{(\alpha\bar{h})^2 - (\lambda_{so}/2)^2}$, $\alpha = \sqrt{2} H_{c2}^{orb}(0)/H_P$, $\psi$ is the digamma function, and $\lambda_{so}$ is the spin–orbit scattering constant. With fixed $H_{c2}^{orb}(0)$, finite $\alpha$ reduces $H_{c2}(0)$, but it recovers toward the original value with increasing $\lambda_{so}$, since spin–orbit scattering suppresses the Pauli paramagnetic effect.

$H_{c2}^{orb}(0)$ was estimated from the initial slope of $H_{c2}(T)$ since the Pauli paramagnetic effect is not relevant near $T_c$ (Supplementary Note 1; Supplementary Fig. 2; and Supplementary Table 1). Then, $H_{c2}(T)$ curves were fitted with $H_P$ and $\lambda_{so}$ as fitting parameters. As shown by the solid lines in Fig. 3a,b, $H_{c2}(T)$ curves are well described by equation (1). Figure 3c shows $H_{c2}(T)$ normalized by $T_c|dH_{c2}/dT|_{T=T_c}$ as a function of $T/T_c$. The normalized $H_{c2}(T)$ curves collapse into a single curve except for Na$_2$CsC$_{60}$, implying that the parameters $\alpha$ and $\lambda_{so}$ are unchanged in a wide $V$ region of the phase diagram, resulting in ($\alpha$, $\lambda_{so}$) = (1.5, 4.4). Figure 3d displays the evolution of $H_{c2}(0)$ as a function of $V$, together with $H_P^{BCS}$. $H_{c2}(0)$ reaches as high as 88 T in Rb$_x$Cs$_{3-x}$C$_{60}$ with $x \lesssim 1$ ($V \gtrsim V_{max}$) very close to the Mott transition. Moreover, $H_{c2}(0)$ is clearly larger than $H_P^{BCS}$ at $V > V_{max}$, and the difference between $H_{c2}(0)$ and $H_P^{BCS}$ becomes pronounced with increasing $V$, although $T_c$ is almost unchanged near the Mott transition.

## Discussion

$H_{c2}(0)$ values reaching $\sim 90$ T are remarkably high for 3D materials. Typical examples of 3D superconductors are cubic Nb$_3$Sn ($H_{c2}(0) = 30$ T, $T_c = 18$ K), which is well known as a material for a SC magnet[18], and Ba$_{1-x}$K$_x$BiO$_3$ ($H_{c2}(0) = 32$ T, $T_c = 28$ K)[19]. MgB$_2$ exhibits strong anisotropy ($H_{c2}(0) = 49$ T and 34 T parallel to the $ab$ plane and $c$ axis, respectively, $T_c = 39$ K)[18] due to its anisotropic electronic structure. $H_{c2}(0)$ of the fullerides is even higher than that of recently discovered H$_3$S super-conductors with likely a cubic structure ($H_{c2}(0) \approx 70$ T, $T_c = 203$ K)[20] despite its much higher $T_c$. In 2D systems under in-plane applied fields, the orbital effect is quenched and higher $H_{c2}$ can be expected. Very large $H_{c2}$ compared with low $T_c$ has been demonstrated in ion-gated MoS$_2$ ($H_{c2}(0) = 52$ T, $T_c = 9.7$ K)[21,22] and monolayer NbSe$_2$ ($H_{c2}(0) = 32$ T, $T_c = 3.0$ K)[23]. In the bulk materials, the in-plane $H_{c2}$ of the cuprates is exceptionally high at above 100 T. However, $H_{c2}$ is no longer a thermodynamic transition line, but a crossover line due to thermal fluctuations. Contrastingly, $H_{c2}$ in pnictides with $T_c \simeq 30$ K is as large as that of fullerides[24]. Therefore, our results highlight the uniquely high $H_{c2}$ measured in the fulleride superconductors that are cubic, and thus, 3D.

To understand the underlying mechanisms for the evolution of $H_{c2}(0)$, we estimated unknown parameters that determine $H_P$ and $H_{c2}^{orb}$ (Supplementary Fig. 1), that is, $\Delta_0$ and the product of parameters in the normal state $m^*/\ell N^{1/3}$. $\Delta_0$ can be

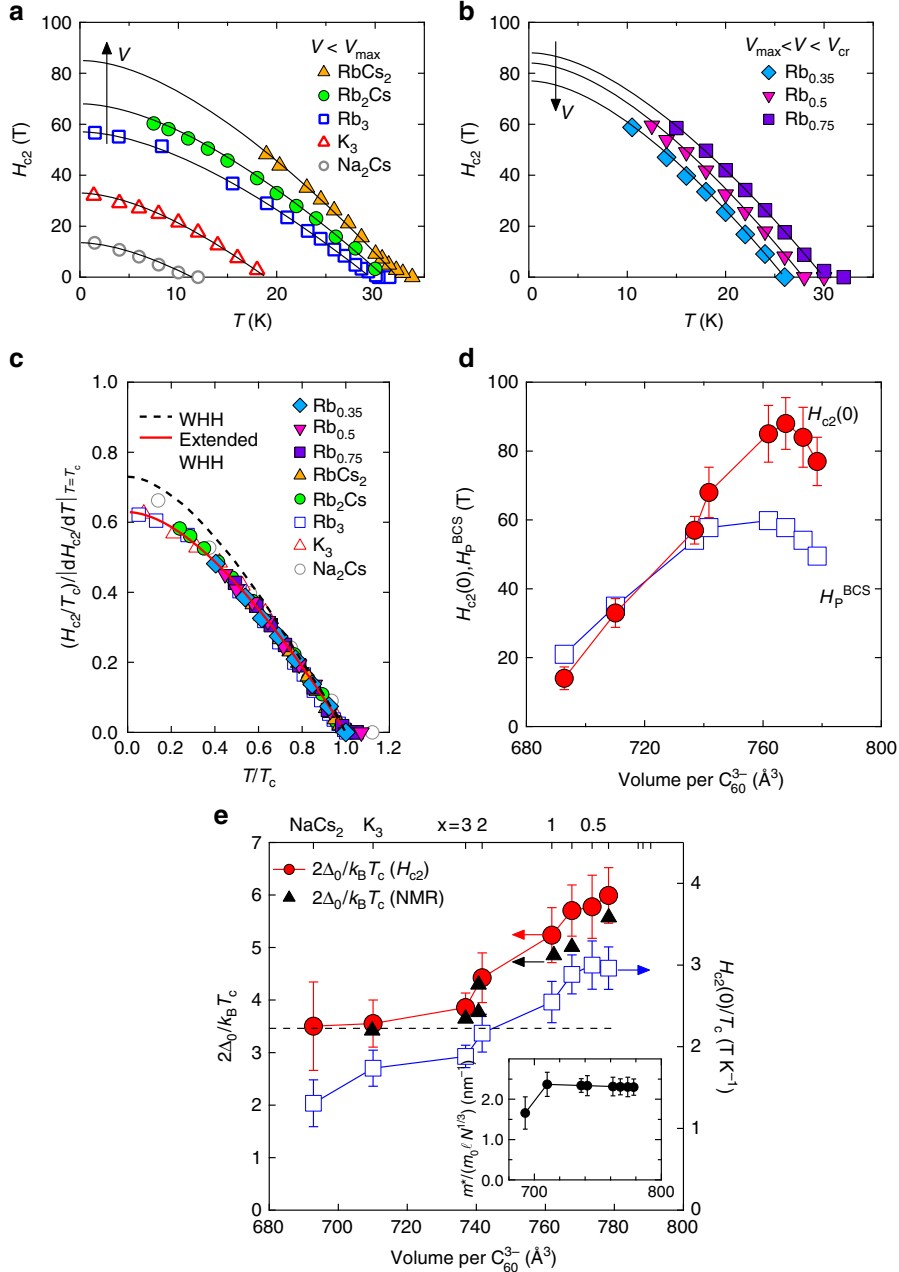

**Figure 3 | Upper critical field in fullerene superconductors.** Temperature dependence of the upper critical field for (**a**) $V \lesssim V_{max}$ and for (**b**) $V_{max} < V < V_{cr}$. The solid lines represent fits using equation (1). (**c**) Normalized $H_{c2}$, $H_{c2}/(T_c|dH_{c2}/dT|_{T=T_c})$, as a function of normalized temperature $T/T_c$. The solid and dashed lines represent calculated $H_{c2}(T)$ using the extended WHH formula (equation (1)) with $\alpha = 1.5$ and $\lambda_{so} = 4.4$ and the conventional WHH formula in the dirty limit, respectively. (**d**) $H_{c2}(0)$, obtained from fits using equation (1), are plotted as a function of volume per $C_{60}^{3-}$ anions. Error bars represent s.d. of the fit to $H_{c2}(T)$ curves. Conventional BCS values of the Pauli limiting field $H_P^{BCS}$ are also shown. (**e**) Evolution of $2\Delta_0/k_BT_c$ and $H_{c2}(0)/T_c$ with approaching to the Mott transition. Error bars on $2\Delta_0/k_BT_c$ and $H_{c2}(0)/T_c$ are calculated from the s.d. in the values of $H_{c2}(0)$ estimated from the least-squares fits of equation (1) to $H_{c2}(T)$ data. $2\Delta_0/k_BT_c$ obtained from NMR measurements is taken from ref. 9. Inset shows $V$ dependence of $m^*/m_0\ell N^{1/3}$ derived using $H_{c2}^{orb}(0)$, $\Delta_0$, and $V$. Error bars are calculated from the s.d. in the values of $H_{c2}(0)$ estimated from the least-squares fits of equation (1) to $H_{c2}(T)$ data..

estimated from $H_P$. In Fig. 3e, the $V$ dependences of $2\Delta_0/k_BT_c$, which is related to the strength of the pairing interaction, and $m^*/m_0\ell N^{1/3}$ ($m_0$ is the bare electron mass) are shown. At low $V$, $2\Delta_0/k_BT_c$ is comparable to the BCS weak-coupling limit value of 3.52. In contrast with the dome-shaped $T_c$, $2\Delta_0/k_BT_c$ continuously increases with increasing $V$ and reaches values as large as 6, indicating a crossover from weak- to strong-coupling super-conductivity on approaching the Mott transition. This is in good agreement with the previous nuclear magnetic resonance results for $Rb_xCs_{3-x}C_{60}$ at ambient pressure[9] and for both

fcc- and A15-$Cs_3C_{60}$ under pressure[25,26], implying universal behaviour in the fullerides. On the other hand, $m^*/m_0\ell N^{1/3}$ is almost constant, indicating that both $H_P/T_c$ and $H_{c2}^{orb}(0)/T_c$ are solely proportional to $2\Delta_0/k_BT_c$. These results lead to the conclusion that the enhancement of $H_{c2}(0)$ is dominated by the strong-coupling effect developing near the Mott transition.

We here recall $H_{c2}(0)$ of other families of high-$T_c$ or strongly correlated superconductors, i.e., cuprates, organic $\kappa$-(ET)$_2$X, and pnictides[24,27–31], having a dome-like SC phase and a proximate antiferromagnetic phase. In Fig. 4, $H_{c2}(0)/T_c$ is displayed as a

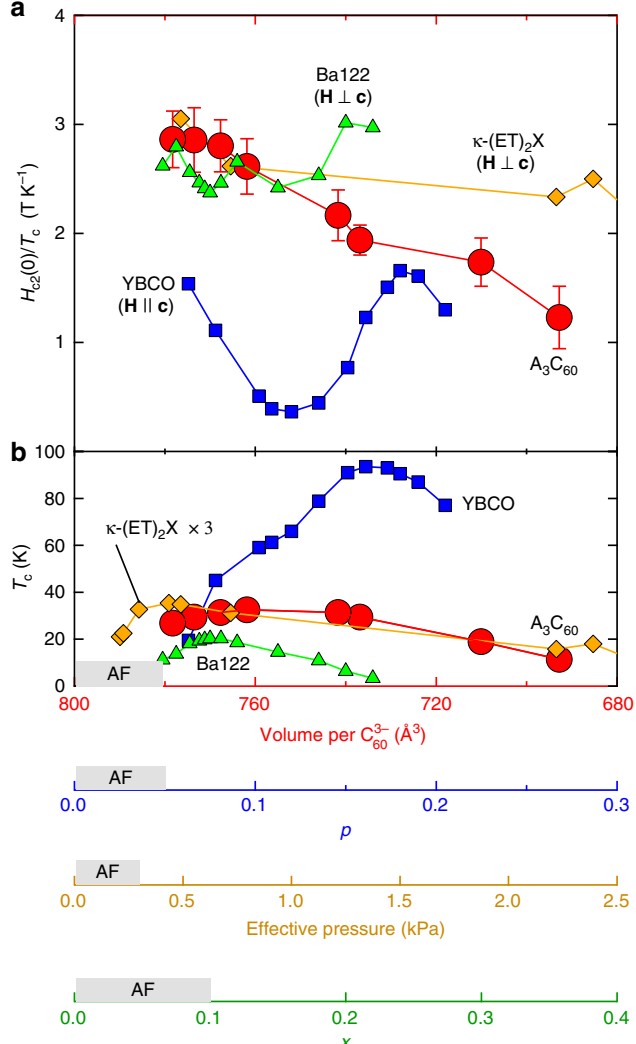

**Figure 4 | Comparison of upper critical field and $T_c$ as a function of control parameters in high-$T_c$ superconductors.** Variations of (**a**) $H_{c2}(0)/T_c$ and (**b**) $T_c$ in high-$T_c$ superconductors, including fullerides $A_3C_{60}$, cuprates $YBa_2Cu_3O_y$ (YBCO) (ref. 27), iron-pnictides $BaFe_{2-x}Ni_xAs_2$ (ref. 24), and organic charge-transfer salts $\kappa$-(BEDT-TTF)$_2$X (X = Cu(NCS)$_2$ and Cu[N(CN)$_2$]Br) (refs 28–30), plotted as a function of control parameters, i.e., lattice volume per $C_{60}^{3-}$ (V), hole concentration (p), Ni content (x), and effective pressure measured from $\kappa$-(BEDT-TTF)$_2$Cu[N(CN)$_2$]Cl (ref. 31). Error bars on $H_{c2}(0)/T_c$ for fullerides represent the s.d. in the values of $H_{c2}(0)$ estimated from the least-squares fits of equation (1) to $H_{c2}(T)$ data.

function of the relevant tuning parameter for each materials family. We show $H_{c2}(0)$ for the in-plane field (H $\perp$ c), where the Pauli paramagnetic effect is dominating, in $\kappa$-(ET)$_2$X and pnictides but show $H_{c2}(0)$ for the out-of-plane field (H $\parallel$ c) in cuprates since there are no reliable estimates of $H_{c2}(0)$ for H $\perp$ c. A remarkable feature of the fullerides is that $H_{c2}(0)/T_c$ appears to be strongly enhanced at $x \leq 1$, where the JT metal phase emerges (Fig. 1c), with retaining nearly optimal $T_c$ and $H_{c2}(0)$ values near the Mott transition. This is in marked contrast to the pnictides and cuprates. In the pnictides, $H_{c2}(0)/T_c$ is almost constant across the SC dome. This is ascribed to the variation of $\Delta_0$, which linearly scales with $T_c$ (ref. 32), implying constant coupling strength. Moreover, in pnictides, $T_c$ and $H_{c2}(0)$ are strongly reduced upon decreasing doping, associated with the appearance of the antiferromagnetic phase. Non-monotonic behaviour in cuprates appears with mass enhancement near $p = 0.08$ and 0.18,

which originates from phase competition between superconductivity and Fermi-surface reconstruction or charge-density-wave order[27]. This is distinct from the continuous evolution of $H_{c2}^{orb}(0)$ in the fullerides (Supplementary Fig. 2), suggesting the absence of such competing states. In $\kappa$-(ET)$_2$X, there is no competing phase near the Mott transition and the molecular degrees of freedom are not relevant to the superconductivity in contrast to the fullerides. Moreover, the SC pairing is most likely mediated by purely electronic interaction, in contrast to the fullerides, where there is considerable controversy because of comparable energy scales in the electron–phonon and electron-electron interactions[33,34]. $\kappa$-(ET)$_2$X shows qualitatively similar behaviour with the strong-coupling effects near the antiferromagnetic phase[35]. However, the enhancement of $H_{c2}(0)/T_c$ is much weaker than that in the fullerides. Therefore, the steep enhancement of $H_{c2}(0)/T_c$ and $2\Delta_0/k_BT_c$ upon entering the JT metal phase cannot be explained solely by the electron correlation effect, highlighting the uniqueness of fullerides among the high-$T_c$ or strongly correlated superconductors. We also emphasize that it is difficult to reconcile the strong-coupling effect with the electron–phonon coupling alone[25]. Our results establish the importance of both molecular characteristics, absent in the atom-based superconductors, involving the dynamical JT effect and the resulting renormalization of the electronic structure and electron correlation effects for both the high-$T_c$ and the high-$H_{c2}$ in the fullerides, as supported by the recent theoretical calculations[33]. This provides a new perspective on realizing robust superconductivity with high $T_c$ and $H_{c2}$ in molecular materials.

## Methods

**Sample synthesis and characterization.** Fullerene superconductors $Na_2CsC_{60}$, $K_3C_{60}$, and $Rb_xCs_{3-x}C_{60}$ ($0 < x \leq 3$) were synthesized by solid-vapor reaction method as described in ref. 9. The samples used here were identical to those in ref. 9. For $Rb_xCs_{3-x}C_{60}$ with $x = 0.5$, 1, and 2, our samples correspond to $Rb_{0.5}Cs_{2.5}C_{60}$ (Sample I), $RbCs_2C_{60}$ (Sample I), and $Rb_2CsC_{60}$ (Sample II) in ref. 9, respectively. The samples were characterized by synchrotron X-ray powder diffraction and magnetization measurements. The phase fraction of the fcc phase was larger than 70% and typical shielding fraction was ~90%.

**Measurements of $H_{c2}$.** Contactless radiofrequency (r.f.) penetration depth measurements were performed using a proximity detector oscillator technique[13] and a pulsed magnetic field up to 62 T in Los Alamos NHMFL. The typical resonant frequency was ~28 MHz. The r.f. technique is highly sensitive to small changes (approximately 1–5 nm) in the r.f. penetration depth $\lambda$, and thus, it is an accurate method for determining $H_{c2}$ of superconductors. Powder samples were compressed into pellets and sealed in thin glass capillaries with a small amount of He gas. Coils that generate and detect microwave signals are directly wound around the capillary (inset of Fig. 2a). The relative change of $\lambda$ is proportional to the relative change of the resonating frequency $f$ through the inductance of the coil, that is, $\Delta f/f \propto \Delta\lambda/\lambda$ (ref. 13). Upper critical field $H_{c2}$ was determined from the field dependence of the frequency shift $\Delta f$ (Supplementary Fig. 1) as the point at which the slope of the r.f. signal in the superconducting state intercepts the slope of the normal state background.

**Data availability.** The data that support the findings of this study are available on request from the corresponding authors (Y.K. or Y.I.).

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

## Acknowledgements

We thank Y. Nomura, R. Arita and E. Tosatti for fruitful discussions. This work was supported in part by Grants-in-Aid for Specially Promoted Research (No 25000003), for Young Scientists (B) (No 2474022), and for Scientific Research on Innovative Areas '3D Active-Site Science' (No 26105004) and 'J-Physics' (No 15H05882) from JSPS, Japan, and SICORP-LEMSUPER FP7-NMP-2011-EU-Japan project (No 283214). This work was also supported by the Mitsubishi Foundation and sponsored by the 'World Premier International (WPI) Research Center Initiative for Atoms, Molecules and Materials,' Ministry of Education, Culture, Sports, Science, and Technology (MEXT) of Japan. K.P. and M.J.R. thank EPSRC for support (EP/K027255 and EP/K027212). M.J.R. is a Royal Society Research Professor. RMcD acknowledges support from U.S. Department of Energy Office of Basic Energy Sciences 'Science at 100 T' program and that a portion of this work was performed at the National High Magnetic Field Laboratory, which is supported by National Science Foundation Cooperative Agreement No DMR-1157490 and the State of Florida.

## Author contributions

Y.K. and Y.I. conceived the experiments. Samples were grown and characterized by R.H.Z., Ya. T., R.H.C., M.J.R. and K.P., and prepared for the measurements of $H_{c2}$ by Y.K. and Yu. T. Pulse-field experiments were performed by Y.K. with help of R.D.M. in National High Magnetic Field Laboratory, Pulsed Field Facility, in Los Alamos National Laboratory, USA. Y.K., K.P. and Y.I. led physical discussions. Y.K. mainly wrote the manuscript.

## Additional information

**Competing financial interests:** The authors declare no competing financial interests.

**Publisher's note**: 

