## [Peer Review File · Nature Communications]

Reviewers' comments:

Reviewer #1 (Remarks to the Author):

This paper reports important new results, and so should certainly be published expeditiously in Nature Communications. H_{c2} is a fundamental characteristic of the SC phase, so from a fundamental physics point of view, knowing its evolution as a function of unit-cell volume, V , and temperature, T , is extremely important. Naturally, H_{c2} is also of major importance for any practical use of a superconductor, which makes the paper doubly interesting.

However, I think the presentation could be improved by removing some of the theoretical discussion - which I think is largely extraneous - and focussing in more closely on making clear and explicit the central new points. I will try to explain what I have in mind, but first let me make it clear that I think the results are so interesting that the authors have "earned" the right to present them any way they want, so these suggestions should be viewed as just SUGGESTIONS, not requirements for acceptance.

Let us start with Fig. 1c. This summarizes much of the previous work of the authors and others and it is important as a point of orientation. But the meaning of the various things are not well and explicitly explained. What is the grey line bounding the Mott insulating regime? Is it a line of first order transitions or is it a crossover line? This should be clearly stated, and the criterion for drawing it should be stated. In particular, the points marked by + are never defined - or if they are I missed it. Indeed, since the "Mott Insulator," in the sense used here, is not a distinct phase of matter, it needs to be defined by some criterion. It would be great if the precise criterion used could be stated. The JT metal is likewise a crossover phenomenon - one that I know the authors are very fond of. I am not sure I am convinced that it is a useful notion, but useful or not it should be sharply defined. Suppose you show me the requisite piece of experimental data - I should be told how to look at that piece of data and tell whether I am looking at a characteristic measurement of the metal regime of the JT metal regime. The sharper the definition, the happier I would be. I think I know what the definition of AFI is in this phase diagram, but even that is not stated and should be.

Still on Fig. 1c, let me make one more suggestion. Phase diagrams are very useful ways of summarizing knowledge. I think it would be good to add the new data to this figure. What I would suggest is having a curve showing $H_{c2}(0)$ on the same figure, with a scale shown on the right edge. Then this one figure would contain the principle take-away message of the paper plus a summary of what is known from before.

On the analysis - I am not exactly sure why it is obvious that BCS theory in the "dirty limit" is what we should be thinking about. Indeed, I would suggest that, in order to incorporate some of the explicit discussion I suggested, cut out much of the standard BCS related discussion and include a good reference instead. Both H_{c2}^{orb} and H_P^{BCS} can be defined without justifying the details $H_{c2}^{\text{orb}} \equiv 0.69 T_c H^{\prime}_c$ and $H_P^{\text{BCS}} \equiv 1.84 T_c$. (By the way, I would suggest a better symbol for this - how about just H_P for "Pauli limit" and also in this defining equation it should be said that this

is for T_c measured in K and H_P in Tesla.)

There are a few other features of the SC state that would be great to mention if the data exists.

Firstly, it would be nice to have a more direct (spectroscopic) measure of Δ_0 either from tunneling or from optics. Does such a thing exist?

Secondly, it would be wonderful to have information about the superfluid stiffness - especially at $T \rightarrow 0$. Does this exist? Is there any sense that the high inferred values of $2\Delta/T_c$ could be associated with a growing importance of phase fluctuations, or is this simply a peculiar strong-coupling limit of BCS theory.

Finally, let me put in a complaint - this is my own perspective and out of step with most people in the field, so the authors should feel free to neglect this. I think that many of the battles in the field are based on false dichotomies. For instance, in the present problem, there can be no doubt that both strong correlations and electron-phonon coupling play a significant role in this. Let me illustrate this by reference to the published literature - a reference I include NOT to urge the authors to reference it, but that this is a place that makes the general point clear. In PRL 69, 212-212 (1992), the issue of the self-consistency of a pure Eliashberg-based treatment of the problem was considered, and it was shown that because the bands in C_{60} are so narrow, that there is not sufficient retardation to permit an effective attraction to emerge from a problem with net repulsive interactions - in other words, a model that ignores strong correlation effects is internally inconsistent. Similarly, the strong renormalization of phonon JT phonon modes proves that they are coupled to the interesting electronic degrees of freedom. There are surely multi-band effects. There is no doubt that there are strong coupling effects, but the fact that the chemical potential does not move from the band center (as best I understand) proves that system is nowhere near the real-space pairing (on-molecule pairs) limit. The issue should not always be framed as a sharp either-or. The issue is what is the simplest context to obtain a qualitative understanding of the basic physics. For instance, even if the electron-phonon coupling and JT distortion play a quantitative role, can one understand the basic phenomena ignoring this? Or conversely, even though the usual pseudo-potential analysis of the Coulomb interactions is clearly flawed, if one simply considered a model in which the Coulomb repulsions were set to be small, would this lead to qualitative predictions that can be falsified. (I think they can be, but this is how I would frame the discussion.)

Reviewer #2 (Remarks to the Author):

The authors present an experimental and theoretical study of the temperature dependence of the upper critical field H_{c2} in superconducting fullerenes. $H_{c2}(T)$ was measured in different fullerenes by a radio-frequency technique in a pulsed magnetic field up to 62T and the T dependence was extrapolated to $T=0$ following the WHH theory and assuming the dirty limit. The values found (up to 90T) are among the highest found for 3D superconductors. In particular, the study of $RbxCs_{3-x}C_{60}$ systems which realize superconductivity near the Mott transition, showed that electron correlations appear to strengthen the superconducting state enhancing the coupling, T_c and H_{c2} .

The article is interesting, the achieved results are sound and clearly presented. As such it is suitable for

publication in Nature Comm. However an issue must be clarified by the authors: the compounds displaying the most interesting behaviour are the fullerenes $Rb_xCs_{3-x}C_{60}$ which are located in the highly correlated JT metal region and which display the high value of $H_{c2}(0)$ and consequently a quite small coherence length, which is, as remarked in the manuscript, of the order of the lattice parameter. However their fractional stoichiometry creates local compositional inhomogeneities which, if on a length-scale larger or comparable with the coherence length, as in this case, can deeply affect the superconducting phase. The influence of this structural disorder has been ignored by the authors. Therefore they should suitably justify their procedure and show that this disorder is not expected to affect their results.

After this issue has been duly clarified in the manuscript I can recommend it for publication in Nature Comm.

REVIEWERS' COMMENTS:

Reviewer #1 (Remarks to the Author):

I like to think that the changes made in response to my suggestions helped improve it, but the main thing is that I do not want (and did not want to) delay publication of beautiful and important results. Congratulations on a lovely paper.

Reviewer #2 (Remarks to the Author):

I consider the authors' explanation satisfactory so I recommend the revised version of the manuscript for publication on Nature Communications.

=====
Response to Reviewer #1:
=====

This paper reports important new results, and so should certainly be published expeditiously in Nature Communications. H_{c2} is a fundamental characteristic of the SC phase, so from a fundamental physics point of view, knowing its evolution as a function of unit-cell volume, V , and temperature, T , is extremely important. Naturally, H_{c2} is also of major importance for any practical use of a superconductor, which makes the paper doubly interesting.

We thank Reviewer #1 for the in depth review and constructive comments that are highly encouraging for our study. Our reply to the questions is the following:

However, I think the presentation could be improved by removing some of the theoretical discussion - which I think is largely extraneous - and focusing in more closely on making clear and explicit the central new points. I will try to explain what I have in mind, but first let me make it clear that I think the results are so interesting that the authors have "earned" the right to present them any way they want, so these suggestions should be viewed as just SUGGESTIONS, not requirements for acceptance.

We agree that the theoretical discussions are largely extraneous. This is a very helpful suggestion for improving our manuscript.

We removed the theoretical discussions in the last paragraph of the original manuscript. Then, we explicitly stated the central new points of this work, i.e., our results imply that the presence of both molecular characteristics that is absent in the atom-based superconductors, involving the dynamical Jahn-Teller effect and resulting renormalization of electronic structure, and electron correlation effects play a significant role for both high- T_c and

high- H_{c2} in the fullerenes. We will explain this point later as a response to the last comments by the referee.

Let us start with Fig. 1c. This summarizes much of the previous work of the authors and others and it is important as a point of orientation. But the meaning of the various things are not well and explicitly explained. What is the grey line bounding the Mott insulating regime? Is it a line of first order transitions or is it a crossover line? This should be clearly stated, and the criterion for drawing it should be stated. In particular, the points marked by + are never defined - or if they are I missed it. Indeed, since the "Mott Insulator," in the sense used here, is not a distinct phase of matter, it needs to be defined by some criterion. It would be great if the precise criterion used could be stated. The JT metal is likewise a crossover phenomenon - one that I know the authors are very fond of. I am not sure I am convinced that it is a useful notion, but useful or not it should be sharply defined. Suppose you show me the requisite piece of experimental data – I should be told how to look at that piece of data and tell whether I am looking at a characteristic measurement of the metal regime of the JT metal regime. The sharper the definition, the happier I would be. I think I know what the definition of AFI is in this phase diagram, but even that is not stated and should be.

We agree that the explanation of Fig. 1c is largely missing. In accordance with the Reviewer #1's comments, we revised Fig. 1c and its caption.

The grey line in Fig. 1c represents the crossover line, not a line of phase transition, from Mott-Jahn-Teller insulator (MJTI) to Jahn-teller (JT) metal (We changed "Mott insulator" to MJTI. We will explain the definition of MJTI later). We explicitly mentioned this in the caption of Fig. 1c in the revised manuscript.

The points marked by + are the crossover temperature obtained from the X-ray powder diffraction, nuclear magnetic resonance spectroscopy, and infrared spectroscopy (Ref. 9).

We defined MJTI as an electron-correlation-driven insulating state

accompanied by the intramolecular dynamic JT effect distorting the C_{60}^{3-} anions and stabilizing the low-spin ($S = 1/2$) states that give rise to an antiferromagnetic insulating (AFI) state at low temperatures. JT metal is defined as a metallic state where the dynamical JT distortions persist.

Before change:

(c) Electronic phase diagram of fcc fullerides. In the metallic regime, gradient shading from green to orange schematically illustrates the conventional metal to Jahn-Teller metal crossover.

After change:

(c) Electronic phase diagram of cubic fullerides. Circles and squares are the superconducting transition temperature T_c for f.c.c. C_{60}^{3-} anion packings with $Pa\bar{3}$ symmetry and $Fm\bar{3}m$ symmetry, respectively. At large V , electron-correlation-driven insulating state (Mott-Jahn-Teller insulator, MJTI) appears, which is accompanied by the intramolecular dynamic Jahn-Teller (JT) effect distorting the C_{60}^{3-} anions and stabilizing the low-spin ($S = 1/2$) states that give rise to an antiferromagnetic insulating (AFI) state at low temperatures. In the metallic regime, gradient shading from green to orange schematically illustrates a crossover from the conventional metal to unusual metallic state where JT distortions persist, which we define as Jahn-Teller metal (JTM) state. The grey line represents the MJTI-to-JTM crossover line, where the crossover temperature (crosses) was obtained from X-ray powder diffraction, nuclear magnetic resonance spectroscopy, and infrared spectroscopy [9]. The ratio of upper critical field at $T = 0$ and T_c , $H_{c2}(0)/T_c$ (yellow triangles), shows an enhancement in the JTM regime.

Still on Fig. 1c, let me make one more suggestion. Phase diagrams are very useful ways of summarizing knowledge. I think it would be good to add the new data to this figure. What I would suggest is having a curve showing $H_{c2}(0)$ on the same figure, with a scale shown on the right edge. Then this one figure would contain the principle take-away message of the paper plus a summary of what is known from before.

We completely agree with the ideas put forward for improving Fig. 1c. We revised Fig. 1c accordingly.

On the analysis - I am not exactly sure why it is obvious that BCS theory in the "dirty limit" is what we should be thinking about.

The mean free path estimated from the transport and optical measurements is as small as, or even smaller than, the intermolecular distance. Therefore, the mean free path is much smaller than the estimated GL coherence length, indicating that the fullerenes are in the dirty limit. We explicitly mention this in the revised manuscript.

In the second paragraph of "Results" section, we added sentences as below.

"It should be noted that the fullerene superconductors are in the dirty limit, $\ell \lesssim \xi_0$ (ℓ and ξ_0 are the mean free path and Pippard coherence length, respectively), as demonstrated by transport and optical measurements [16,17]. The orientational disorder of the C_{60}^{-3} anions can account for the short ℓ , which is comparable to the intermolecular separation."

Indeed, I would suggest that, in order to incorporate some of the explicit discussion I suggested, cut out much of the standard BCS related discussion and include a good reference instead. Both H_{c2}^{orb} and H_P^{BCS} can be defined without justifying the details $H_{c2}^{orb} \approx 0.69 T_c H'_{c2}$ and $H_P^{BCS} \approx 1.84 T_c$. (By the way, I would suggest a better symbol for this - how about just H_P for "Pauli limit" and also in this defining equation it should be said that this is for T_c measured in K and H_P in Tesla.)

We agree the referee's suggestions. We revised the standard BCS related discussion and cited references.

There are a few other features of the SC state that would be great to mention if the data exists. Firstly, it would be nice to have a more direct (spectroscopic) measure of Δ_0 either from tunneling or from optics. Does such a thing exist?

For K_3C_{60} and Rb_3C_{60} , Δ_0 has been measured by spectroscopic experiments, including scanning tunneling microscopy, optical measurements, and photoemission. However, in $Rb_xCs_{3-x}C_{60}$ near the Mott transition, which is the main focus of the present study, such spectroscopic experiments have never been reported.

Secondly, it would be wonderful to have information about the superfluid stiffness - especially at $T \rightarrow 0$. Does this exist? Is there any sense that the high inferred values of $2\Delta_0/T_c$ could be associated with a growing importance of phase fluctuations, or is this simply a peculiar strong-coupling limit of BCS theory.

We agree that this is an important point for discussing the nature of dome-shaped T_c since the superconducting gap Δ_0 and superconducting phase stiffness ρ_s are the primary factors that control T_c . The relation between Δ_0 , ρ_s , and T_c would give information about the underlying physical phenomena on T_c dome, such as BEC-BCS crossover [Uemura, J. Phys.: Condens. Matter 82004]], phase competition, and the superconducting phase fluctuations [Emery & Kivelson, Nature (1995)].

In the fullerides, it has been shown that T_c and ρ_s concomitantly increase with increasing V at low V where $2\Delta_0/k_B T_c$ is almost constant (i.e., Δ_0 varies linearly with T_c), although carrier density is unchanged [Uemura *et al.*, Physica C **235-240**, 2501 (1994)]. However, there has been no experimental report on ρ_s for $Rb_xCs_{3-x}C_{60}$ near the Mott transition. Thus, direct measurement of ρ_s is strongly desired for $Rb_xCs_{3-x}C_{60}$.

Finally, let me put in a complaint - this is my own perspective and out of step with most people in the field, so the authors should feel free to neglect

this. I think that many of the battles in the filed are based on false dichotomies. For instance, in the present problem, there can be no doubt that both strong correlations and electron-phonon coupling play a significant role in this. Let me illustrate this by reference to the published literature - a reference I include NOT to urge the authors to reference it, but that this is a place that makes the general point clear. In PRL 69, 212-212 (1992), the issue of the self-consistency of a pure Eliashberg-based treatment of the problem was considered, and it was shown that because the bands in C₆₀ are so narrow, that there is not sufficient retardation to permit an effective attraction to emerge from a problem with net repulsive interactions - in other words, a model that ignores strong correlation effects is internally inconsistent. Similarly, the strong renormalization of phonon JT phonon modes proves that they are coupled to the interesting electronic degrees of freedom. There are surely multi-band effects. There is no doubt that there are strong coupling effects, but the fact that the chemical potential does not move from the band center (as best I understand) proves that system is nowhere near the real-space pairing (on-molecule pairs) limit. The issue should not always be framed as a sharp either-or. The issue is what is the simplest context to obtain a qualitative understanding of the basic physics. For instance, even if the electron-phonon coupling and JT distortion play a quantitative role, can one understand the basic phenomena ignoring this? Or conversely, even though the usual pseudo-potential analysis of the Coulomb interactions is clearly flawed, if one simply considered a model in which the Coulomb repulsions were set to be small, would this lead to qualitative predictions that can be falsified. (I think they can be, but this is how I would frame the discussion.)

As the referee pointed out, we should take into account both the electron-phonon coupling and electron correlations. Indeed, the observed strong-coupling effect cannot be explained solely by the electron-phonon coupling or electron-electron interactions. This is explained as follows.

First, in the scenario of electron-phonon coupling, intermolecular phonons

(with energy $\sim 100 \text{ cm}^{-1}$) are required as a pairing interaction to account for very large $2\Delta_0/k_B T_c$ up to 6 at large V . On the other hand, the weak-coupling value ($2\Delta_0/k_B T_c \sim 3.5$) can be explained only by intramolecular phonons, involving JT phonon modes (with energy $\sim 1000\text{-}1500 \text{ cm}^{-1}$). Therefore, large enhancement of $2\Delta_0/k_B T_c$ requires a crossover of the distinct phonon modes that are active for the pairing. This is highly unlikely since the intramolecular phonon modes are always present over the phase diagram.

Second, the analysis of the previous specific heat measurements (ref.9 in the manuscript) demonstrate that strong enhancement of the specific heat jump $\Delta C/T_c$ cannot be explained by a weak-coupling model, in which the intramolecular phonon modes and the density of states obtained from the DFT band structure calculations are assumed. This suggests that, to account for the strong-coupling effect, enhancement of the pairing interaction or density of states (DOS) is required. According to theoretical calculations (ref.36 in the revised manuscript), electron-phonon interactions rather decrease with increasing V , suggesting that DOS is enhanced in the JT metal region. It is natural to consider that the electronic structure is strongly renormalized by the electron correlation effects as the Mott transition is approached. Since the on-site electron-electron repulsion is comparable with the bandwidth in the fullerenes, electron correlation always plays a significant role, suggesting that the presence of both the dynamical JT effect and electron correlation is important for the strong-coupling effect.

Therefore, the observed steep increase of $H_{c2}(0)/T_c$ and strong-coupling effect upon entering the JT metal phase implies that cooperative interplay between the molecular properties of C_{60}^{3-} anions and electron correlations play a significant role for realizing both high- T_c and high- H_{c2} in the fullerenes. We explicitly mention this in the revised manuscript.

=====
Response to Reviewer #2:
=====

The authors present an experimental and theoretical study of the temperature dependence of the upper critical field H_{c2} in superconducting fullerenes. $H_{c2}(T)$ was measured in different fullerenes by a radio-frequency technique in a pulsed magnetic field up to 62T and the T dependence was extrapolated to $T=0$ following the WHH theory and assuming the dirty limit. The values found (up to 90T) are among the highest found for 3D superconductors. In particular, the study of $Rb_xCs_{3-x}C_{60}$ systems which realize superconductivity near the Mott transition, showed that electron correlations appear to strengthen the superconducting state enhancing the coupling, T_c and H_{c2} . The article is interesting, the achieved results are sound and clearly presented. As such it is suitable for publication in Nature Comm.

We thank Reviewer #2 for the in depth review and constructive comments that are highly encouraging for our study. Our reply to the questions is the following:

However an issue must be clarified by the authors: the compounds displaying the most interesting behaviour are the fullerenes $Rb_xCs_{3-x}C_{60}$ which are located in the highly correlated JT metal region and which display the high value of $H_{c2}(0)$ and consequently a quite small coherence length, which is, as remarked in the manuscript, of the order of the lattice parameter. However their fractional stoichiometry creates local compositional inhomogeneities which, if on a length-scale larger or comparable with the coherence length, as in this case, can deeply affect the superconducting phase. The influence of this structural disorder has been ignored by the authors. Therefore they should suitably justify their procedure and show that this disorder is not expected to affect their results.

After this issue has been duly clarified in the manuscript I can recommend it for publication in Nature Comm.

This is a helpful and new perspective which we have included in the paper as detailed below.

Physical and chemical pressure studies on fcc-C₃S₃C₆₀ show essentially the same behaviors in the superconducting properties, including T_c and $2\Delta_0/k_B T_c$ (Refs. 7,9,25). Moreover, pressure studies on A15-C₃S₃C₆₀ and fcc-C₃S₃C₆₀ also exhibits the comparable behaviors to those in chemically-pressurized fcc-C₃S₃C₆₀, i.e., Rb_xC_{3-x}S₃C₆₀, in both the normal state properties, such as the presence of JTM state near the Mott transition, and superconducting properties (Ref. 9,12,25,26). In contrast to Rb_xC_{3-x}S₃C₆₀, A15-C₃S₃C₆₀ is free from both the fractional stoichiometry and orientational disorder of C₆₀⁻³ anions. Therefore, we conclude that fractional stoichiometry does not affect the present results.

On the other hand, the shortest length scale of the structural disorder is most likely determined by the orientational disorder (merohedral disorder) of C₆₀⁻³ anions (Potocnik *et al.*, Chem. Sci. 5, 3008 (2014)). There are two equal C₆₀⁻³ orientations, and they are randomly and equally distributed at low temperatures. The orientational disorder is present even in the stoichiometric compounds such as K₃C₆₀. The length scale of the orientational disorder is roughly estimated to be of the order of the intermolecular distance (about 9 Å) or intermolecular C-C spacing (about 3 Å). This is consistent with the mean free path ℓ estimated from the transport and optical measurements, where ℓ is found to be smaller than the estimated GL coherence length. We explicitly mention this in the revised manuscript.

The length scale of structural disorder in the alkali-metal sites is estimated as follows. In the fcc structure of the fullerides, there are tetrahedral and octahedral sites for the alkali atoms at (1/4,1/4,1/4) and (1/2,0,0), respectively. In Rb_xC_{3-x}S₃C₆₀ with $x < 1$, the octahedral sites are occupied only by Rb atoms and the tetrahedral sites by both Rb and Cs atoms. Therefore, the length scale of the fractional stoichiometry is comparable to the distance of the

tetrahedral site, which is about 10 Å. This is also comparable to the intermolecular distance and is smaller than the GL coherence length.

In the second paragraph of “Results” section, we added sentences as below.

“It should be noted that the fulleride superconductors are in the dirty limit, $\ell \lesssim \xi_0$ (ℓ and ξ_0 are the mean free path and Pippard coherence length, respectively), as demonstrated by transport and optical measurements [16,17]. The orientational disorder of the C_{60}^{-3} anions can account for the short ℓ , which is comparable to the intermolecular separation.”